# Wound Infection of Snakebite from Venomous *Protobothrops mucrosquamatus*, *Viridovipera stejnegeri* and *Naja atra* in Taiwan: Validation of BITE and Cobra BITE Scoring Systems and their Bacteriological Differences in Wound Cultures

**DOI:** 10.3390/toxins15010078

**Published:** 2023-01-15

**Authors:** Heng Yeh, Shi-Ying Gao, Chih-Chuan Lin

**Affiliations:** 1Department of Emergency Medicine, Lin-Kou Medical Center, Chang Gung Memorial Hospital, Taoyuan 33305, Taiwan; 2School of Medicine, College of Medicine, Chang Gung University, Taoyuan 33302, Taiwan

**Keywords:** wound infections, snakebites, *Protobothrops mucrosquamatus*, *Viridovipera stejnegeri*, *Naja atra*

## Abstract

Patients bitten by *Protobothrops mucrosquamatus*, *Viridovipera stejnegeri*, and *Naja atra* develop different degrees of wound infection. This study validated BITE and Cobra BITE scoring systems that we established previously. Bacteriological studies of patients with wound infection were conducted. The operating characteristic curves and area under the curve (AUC) and wound infection rates were compared between the derivation set (our previous study patient population) and the validation set (new patient cohorts enrolled between June 2017 and May 2021). No significant differences in the AUC for both the BITE (0.84 vs. 0.78, *p* = 0.27) and Cobra BITE (0.88 vs. 0.75, *p* = 0.21) scoring systems were observed between the derivation and validation sets. *Morganella morganii* and *Enterococcus faecalis* were the two most commonly detected bacteria in the microbiological study. More bacterial species were cultured from *N. atra*-infected wounds. Antibiotics such as amoxicillin with clavulanic acid, oxacillin, and ampicillin may not be suitable for treating patients with *P. mucrosquamatus*, *V. stejnegeri*, and *N. atra* bites in Taiwan. Carbapenem, third-generation cephalosporins, and fluoroquinolone may be superior alternatives.

## 1. Introduction

Six clinically important venomous snake species are found in Taiwan, namely *Protobothrops mucrosquamatus* (Taiwan habu), *Viridovipera stejnegeri* (green bamboo viper), *Naja atra* (Taiwan or Chinese cobra), *Bungarus multicinctus* (Taiwan banded krait), *Deinagkistrodon acutus* (hundred pacer viper), and *Daboia siamensis*. Two-thirds of all venomous snakebites in Taiwan are caused by *P. mucrosquamatus* and *V. stejnegeri* [1]. Patients with venomous snakebites present with various symptoms, including swelling, tenderness, and local heat in the affected limb tissue; cellulitis; and wound infection. *N. atra* accounts for approximately one-fifth of poisonous snakebites in Taiwan [1]. Instead of neurological complications, patients bitten by *N. atra* experience tissue damage, wound necrosis, and necrotizing fasciitis because of cytotoxins present in the venom and, these wounds have a high risk of infection [1,2,3,4].

The available treatments for these snakebites are toxoid vaccine injection, antibiotics, symptomatic relief agents, and antivenins [5]. Physicians prescribe freeze-dried hemorrhagic (FH) antivenom to treat bite wounds caused by *P. mucrosquamatus* or *V. stejnegeri* and freeze-dried neurotoxic (FN) antivenom for wounds caused by *N. atra* [6]. Except for wound necrosis caused by *N. atra* bites that require antibiotic treatment, determining when and how to administer antibiotics is challenging for clinicians treating patients with snakebites; this problem therefore warrants investigation. We have previously developed scoring systems for wounds in the BITE [7] and Cobra BITE studies [8]; however, they have not been validated yet. This study validated our previous two scoring systems for wound infections caused by the bites of *P. mucrosquamatus*, *V. stejnegeri*, and *N. atra*. Moreover, we compared bacteriological results between patients bitten by *P. mucrosquamatus* or *V. stejnegeri* and those bitten by *N. atra* bite. Finally, we provide practical antibiotic treatment suggestions to clinical physicians on the basis of bacteriological wound culture results, including antibiotic sensitivity tests.

## 2. Results

### 2.1. Patient Characteristics of the Validation Cohort of the BITE Study

A total of 95 patients received at least one vial of FH antivenom. A median of two vials (interquartile range [IQR] 2–4) of FH antivenom was used for these patients. Among the patients who received FH antivenom treatment, 35 were diagnosed as having wound infections during the treatment course. Lymphocyte count was the only laboratory variable that differed significantly between the patients with and without wound infection (Appendix A). Furthermore, 51 of the 95 patients required hospitalization, and the length of hospitalization was 3.86 ± 3.76 days. The wound infection group had more patients requiring hospital admission compared with the no wound infection group (30/35 vs. 21/60, *p* < 0.01). The length of hospitalization in the wound infection group was significantly longer than that in the no wound infection group (5.31 ± 4.40 vs. 1.79 ± 2.85, *p* < 0.01; Appendix A). Among the hospitalized patients, five received debridement, five received fasciotomy, and two received graft treatment. Two patients received debridement with fasciotomy and graft treatment, and two received debridement and fasciotomy. No patients died, but one patient was admitted to the intensive care unit (ICU) because of the development of rhabdomyolysis and acute kidney injury.

### 2.2. Patient Characteristics of the Validation Cohort of the Cobra BITE Study

Thirty-one patients bitten by *N. atra* received at least one FN treatment vial. Six patients developed wound necrosis and were not enrolled in the Cobra BITE score validation cohort. Eleven of the remaining 25 patients with *N. atra* bites without wound necrosis had wound infections. No significant differences in age, sex, and laboratory variables were noted between the wound infection and no wound infection groups (Appendix A). The median doses of FN were 6 (IQR 2–10) and 4 (IQR 2–6) in the wound infection and no wound infection groups (*p* = 0.23), respectively. The wound infection group had more patients requiring admission compared with the no wound infection group (*n* = 10/11 vs. 7/14, *p* = 0.04). The length of hospitalization in the wound infection group was significantly longer than that in the no wound infection group (6.27 ± 4.38 vs. 1.64 ± 1.86, *p* = 0.01; Appendix A). Among the patients bitten by *N. atra*, six received debridement, four received fasciotomy, and three received graft treatment. Two patients received debridement with fasciotomy and graft treatment, and one received debridement and fasciotomy. One patient received debridement with graft treatment. No patient was admitted to the ICU or died.

### 2.3. Comparison of Patient Characteristics between the Derivation and Validation Cohorts for BITE and Cobra BITE Studies

The samples of the patients in the derivation and validation groups were similar, except that both validation sets had higher infection and wound infection rates (Table 1).

### 2.4. Validation of the BITE and Cobra BITE Studies

The BITE and Cobra BITE scoring systems maintained their predictive ability when applied to their validation sets. The BITE and Cobra BITE scoring systems successfully stratified patients into groups with an increased risk of wound infection (Figure 1 and Figure 2).

According to the BITE scoring system, the wound infection rates in the derivation cohort were 3.03%, 16.35%, 54.65%, and 60.87% for scores of 0, 1, 4, and 5, respectively. The wound infection rates in the validation cohort were 12.50%, 8.33%, 45.45%, and 67.86% for scores of 0, 1, 4, and 5, respectively (Figure 1a). The area under the receiver operating characteristic (ROC) curve of the derivation set was 0.84 (95% confidence interval [CI], 0.80 to 0.88). The area under the ROC curve of the validation set in the BITE study was 0.78 (95% CI, 0.69 to 0.88, Figure 1b). The DeLong test did not indicate significant differences in the AUC between the validation and derivation cohorts (*p* = 0.27).

According to the Cobra BITE scoring system, the wound infection rates in the derivation cohort were 1.92%, 11.11%, 0.00%, 36.00%, and 50% for scores of 0, 1, 3, 4, and 5, respectively. The infection rates were 100% for scores 7 and 8, respectively. The wound infection rates in the validation cohort were 0.00%, 33.33%, 0.00%, 75.00%, and 50% for scores of 0, 1, 3, 4, and 5, respectively. The infection rate was 100% for a score of 8. (Figure 2a). However, the wound infection rate for a score of 4 differed between the derivation and validation cohorts in which the validation cohort had a higher risk of wound infection. This difference might be due to the small number of patients (*n* = 7) with a score of 4 in the validation cohort. The area under the curve (AUC) of the derivation cohort in the Cobra BITE study was 0.88 (95% CI, 0.81 to 0.95), and the AUC of the validation set was 0.75 (95% CI, 0.56 to 0.98; Figure 2b). We compared the AUC between the derivation and validation sets by using the DeLong test and determined that the AUC did not significantly differ between the two sets (*p* = 0.21).

### 2.5. Bacteriology and Antibiotic Susceptibility Testing

The bacteriology and culture reports of wound swabs from *P. mucrosquamatus*, *V. stejnegeri*, and *N. atra* for the derivation and validation sets are presented in Table 2. *Morganella morganii* and *Enterococcus faecalis* were the most common bacteria in wound cultures regardless of the snake species. These two bacteria were also the most prevalent in the wounds of the patients with infection caused by *N. atra* bites.

Generally, more bacteria were isolated from wounds caused by *N. atra* bites. Multiple aerobic Gram-positive bacteria, such as *Staphylococcus* species, *Viridans streptococcus*, and *Corynebacterium* species, were found in the patients with *P. mucrosquamatus* and *V. stejnegeri* bites. Considerably higher numbers of aerobic Gram-negative bacteria, such as *Acinetobacter* species, *Shewanella algae*, *Proteus vulgaris*, *Citrob freundii*, *Bacteroides fragilis*, *Bacteroides thetaiotaomicron*, and *Serratia marcescens*, and anaerobic bacteria were detected in the infected wounds of patients with *N. atra* bites.

### 2.6. Use of Empirical Antibiotics and Antibiotic Therapy for Infected Snakebite Wounds in the Validation Set

Twenty-two patients who received FN antivenin therapy received empirical intravascular antibiotic treatment. Before culture results were obtained, amoxicillin with clavulanic acid was the most commonly used antibiotic, followed by cefuroxime and clindamycin (Table 3). However, the wound culture reports demonstrated that these empirical antibiotics were ineffective in controlling wound infection because the wounds were infected by antimicrobial-resistant bacteria.

Among the 95 patients who received FH antivenin treatment, 38 received intravenous antibiotics in the emergency department (ED). Amoxicillin with clavulanic acid was still the most frequently used antibiotic. Cefazolin with gentamicin, clindamycin, vancomycin with ceftriaxone, oxacillin, ampicillin with sulbactam, and cefuroxime were all used to treat the patients with snakebites. Similarly, the empirical use of amoxicillin with clavulanic acid and cefazolin was not sufficient to control wound infection because of the presence of antibiotic-resistant bacteria.

### 2.7. Validation of BITE and Cobra BITE Studies and the Use of Antibiotics

The findings of the Cobra BITE study indicated that antibiotics and ≥4 doses of antivenin should be administered to patients with *N. atra* bites but without tissue necrosis (Cobra BITE score ≥5, which indicates a wound infection rate of at least 50%). However, if tissue necrosis occurs, these patients should be admitted to the ward and receive antibiotic treatment (Figure 3).

On the basis of the findings of the BITE study, we suggest the administration of antibiotics to patients with a BITE score of 4 or 5 points [7]. When patients are bitten by *P. mucrosquamatus* and *V. stejnegeri* and receive FH antivenin in the ED, antibiotics should be administered if the neutrophil-to-lymphocyte ratio of patients is ≥19.84 (Figure 4).

## 3. Discussion

We validated our two previously reported snakebite-related wound infection prediction scoring systems in this study. In addition, we compared the bacteriology of infected wounds caused by *P. mucrosquamatus*, *V. stejnegeri*, and *N. atra* bites. Although *M. morganii* and *E. faecalis* were the most commonly detected bacteria in wound cultures regardless of the snake species, considerable differences in causative bacterial strains were noted among wound infections caused by *P. mucrosquamatus*, *V. stejnegeri*, and *N. atra* bites. The findings of this study can help physicians in promptly treating snakebite wound infections.

### 3.1. Bacteriology of Snake Oral Cavity and Wound Culture in N. atra Bites

This study examined eight samples from patients’ wound or pus cultures. *M. morganii* and *E. faecalis* were the most frequently detected bacteria in wound or pus cultures. This finding is consistent with those of previous studies [7,8,9]. In 2021, Mao et al. [10] reported that *M. morganii* was the most frequently detected bacterium in both *N. atra* bite wounds and oropharyngeal swabs. However, a study published in 2022 by Chuang et al. [11] indicated that *E. faecalis* was the most commonly isolated bacterium from the oral cavity of *N. atra* and *M. morganii* was detected in only 1 of the 30 wound culture samples collected from patients with *N. atra* bites. *Pseudomonas* species were the most frequently detected aerobic Gram-negative bacteria in *N. atra* wounds.

Shek et al. [12] reported that *M. morganii* was detected in the oral cavity of 23 of 32 *N. atra* in Hong Kong, and *Pseudomonas* species were detected in the oral cavity of only eight snakes. In 2016, Su et al. [13] reported that *M. morganii* and *E. faecalis* were the most commonly detected bacteria in the wound culture of specimens obtained from patients with *N. atra* bites who underwent surgery; however, they did not determine *Pseudomonas* species; this finding is consistent with that of our study. Another study conducted in India reported that *Pseudomonas* and *Proteus* species instead of *M. morganii* were the most widely detected Gram-negative bacteria in the oral cavity of *N. naja*. *Bacillus* and *Staphylococcus* species instead of *E. faecalis* were the most frequently identified Gram-positive bacteria [14]. Further research is warranted because of the discrepancies in these findings and because the literature cannot fully explain the gap in findings between the wound and oral cavity cultures. The immunocompromised state of injuries caused by snakebite wounds might provide a suitable culture medium for these bacteria.

Although the species and habitats of snakes differ, Panda et al. [14] reported that nearly half of the bacterial cultures were resistant to amoxicillin with clavulanic acid, and nearly 100% of bacterial culture reports exhibited resistance to penicillin. Therefore, they did not suggest using amoxicillin with clavulanic acid, oxacillin, and penicillin as prophylactic antibiotics [14]. These results are similar to those reported by Lam et al. [15]. They indicated that only 23% of bacterial cultures were sensitive to amoxicillin with clavulanic acid, and approximately 40% were sensitive to cefuroxime. *Enterobacter* species exhibited only approximately 14% sensitivity to amoxicillin with clavulanic acid. Mao et al. [16] reported that *M. morganii* should be treated with third-generation cephalosporin or fluoroquinolone with aminopenicillin instead of first- and second-generation cephalosporin because of their resistance. A possible cause of resistance to first-generation cephalosporins can be the presence of the VGH116 genome in *M. morganii* [17]. Resiere et al. [18] suggested that amoxicillin with clavulanic acid was not suitable for treating snakebites and that third-generation cephalosporin may be administered if required. Ngoc et al. [19] conducted a study in in Vietnam and examined the wound culture reports of *Naja* species’ snakebites. They observed that *E. faecalis* exhibited sensitivity to all antibiotics and *M. morganii* was sensitive to cephalosporin, quinolone, and carbapenem. They also noted that the treatment of wounds caused by *Naja* species with clindamycin and ciprofloxacin reduced wound necrosis and infection. Considering that *M. morganii* and *E. faecalis* were the most commonly detected bacteria in cultures, we suggest using third-generation cephalosporins (such as ceftriaxone and ceftazidime) and fluoroquinolones (ciprofloxacin, levofloxacin).

### 3.2. Bacteriology of Snake Oral Cavity and Wound Culture in P. mucrosquamatus and V. stejnegeri Bites

*P. mucrosquamatus* and *V. stejnegeri* bites are more common than *N. atra* bites in Taiwan [20]. Unlike *N. atra* bites, these snakebites cause less bacterial wound infection and cellulitis [4]. *Pseudomonas* species were the most frequently detected aerobic Gram-negative bacteria in the oral cavity of *P. mucrosquamatus* and *V. stejnegeri* [11]. *M. morganii* and *E. faecalis* have not been frequently identified in the bacterial culture reports of *V. stejnegeri* bites [21]. Similarly, another study revealed that bites from *P. mucrosquamatus* produced similar results—*M. morganii* and *Enterococcus species* were not the dominant pathogens in isolated culture reports [22]. These findings are consistent with those of both the derivation and validation cohorts of our study. Thus, third-generation cephalosporin, fluoroquinolone, piperacillin with tazobactam, vancomycin, and teicoplanin may still be adequate treatment options.

### 3.3. Antibiotic Use

Because *M. morganii* and *E. faecalis* were the most commonly identified bacteria in cultures, third-generation cephalosporins (such as ceftriaxone and ceftazidime) and fluoroquinolones (ciprofloxacin and levofloxacin) should be administered. However, the microorganisms may exhibit resistance to first- and second-generation cephalosporins, such as cefazolin, cefuroxime, and ampicillin with sulbactam [11].

A more sophisticated approach for the selection of antibiotics should be adopted. Patients may require multiple surgical procedures because of *N. atra* bite–related wound necrosis [23,24]; thus, use of antibiotics such as metronidazole, amoxicillin with clavulanic acid, and piperacillin with tazobactam to combat both aerobic and anaerobic microorganisms is recommended [25] (Figure 3). Patients with *P. mucrosquamatus* or *V. stejnegeri* bites may require antibiotics that can kill anaerobic Gram-positive bacteria (Figure 4).

Prophylactic antibiotics are still extensively used in Taiwan to treat patients with snakebites [26]. In our study, most patients received oral amoxicillin with clavulanic acid as prophylactic antibiotics or intravascular amoxicillin with clavulanic acid as empirical antibiotics following hospital admission. According to the findings of antibiotic susceptibility testing, some patients who underwent FN treatment were not suitable for treatment with cefuroxime, cefazolin, and ampicillin with sulbactam for cellulitis. Resistance may be exhibited by *M. morganii*, *P. vulgaris*, and *E. faecalis*.

### 3.4. Limitations

This study validated the BITE and Cobra BITE scoring systems. However, this study had some limitations. The number of samples collected in this study was limited, particularly for patients with *N. atra* bites, although this study was conducted in the largest hospital in Taiwan.

We calculated the required power by comparing the admission rate between the patients with and without wound infection in the BITE and Cobra BITE validation sets (the power was 0.95 and 0.96, respectively). By using an independent χ² test, we calculated a presumed effect size of 0.9 under a two-tailed comparison with a preset α level of 0.05.

Another limitation was that our central laboratory only tested the antibiotics sensitivity test but lacking the antibiotics concentrations.

## 4. Conclusions

Both our previously published BITE and Cobra BITE scoring systems were successfully validated in this study. These two practical tools may aid physicians in treating patients in Taiwan bitten by *N. atra*, *P. mucrosquamatus*, and *V. stejnegeri.* In addition, some antibiotics such as amoxicillin with clavulanic acid, oxacillin, and ampicillin may no longer be reliable and useful for snakebite wound treatment. Third-generation cephalosporins, carbapenem, and fluoroquinolone can be considered as alternatives. 

## 5. Materials and Methods

### 5.1. Patient Selection, Data Resource Setting, and Collected Variables

This study adopted a retrospective design. We searched the electronic medical records from the ED of the Chang Gung Memorial Hospital (CGMH) network, which contains two medical centers (Linkou and Kaohsiung CGMH) and five regional hospitals (Taoyuan, Taipei, Keelung, Yunlin, and Chiayi CGMH). The CGMH network is the largest private medical network in Taiwan. This medical system contains approximately 10,000 beds and receives approximately 2.4 million admissions annually. The CGMH network also receives approximately 8.2 million outpatients every year. Thus, nearly a third of the Taiwanese population has received treatment in the CGMH network hospitals [27].

The patient cohorts in the BITE and Cobra BITE study were considered the derivation set. We used a derivation set to figure out the possible predictive factors and established the scoring systems. After we developed the two scoring models (the BITE and Cobra BITE study) for the evaluation of snakebite wound infection risk, we still needed to confirm the validity of our previous achievement. Therefore, we collected another separate and independent group of patients bitten by *P. mucrosquamatus*, *V. stejnegeri*, or *N. atra.* These groups were regarded as the validation set. We used the validation set to verify the two scoring models.

We identified adult patients who were bitten by venomous snakes, namely *N. atra*, *P. mucrosquamatus*, and *V. stejnegeri*, and subsequently sought treatment at our ED between 1 June 2017, and 31 May 2021. These patients received one vial of FH antivenom (the designated antivenom for *P. mucrosquamatus* and *V. stejnegeri* snakebites) or FN antivenom (the designated antivenom for *N. atra* bites) at least during their visit to EDs. Patients who received both FH and FN simultaneously during their ED visits or received other antivenoms were excluded. Patients who did not receive any antivenom treatment were also excluded. These patients were included in the validation cohort.

The characteristics of patients who comprised the derivation cohort have been described in our previous studies. Two emergency physicians carefully reviewed these patients’ medical records (C.-C.L. and H.Y). Collected variables were abstracted from the electronic medical record database. Demographic characteristics such as age and sex were recorded. Laboratory variables measured in the ED included white blood cell count with differential count, neutrophil-to-lymphocyte ratio, hemoglobin, red blood cell distribution width, platelet count, prothrombin time, activated partial thromboplastin time, blood creatinine, blood urea nitrogen, alanine aminotransferase, aspartate aminotransferase, myoglobin, potassium, sodium, and blood glucose. Additionally, other details regarding treatment modalities such as antivenom dose, method of antibiotic administration, type of surgical procedure, and hospitalization duration were obtained. The bacteriology data of wound and pus cultures from snakebite victims were retrieved.

### 5.2. Current Treatment Protocol for Patients with Snakebites and Definitions of Wound Infection, Polymicrobial Infection, and Wound Necrosis

Our routine management protocols are based on World Health Organization’s guidelines [28] and have been described in previous articles. The selection of antivenom was based on the venomous snake species. Thus, clinicians would ask patients in the ED to identify the snake by using a pictorial chart available in our EDs, and clinicians would confirm this identification based on the patient’s symptoms and signs. The patients would be monitored for at least 24 h after the administration of appropriate antivenoms either in ED or the ordinary ward. If clinical symptoms, such as limb swelling, improved, the patients could return home and follow-up was conducted at an outpatient clinic. This protocol was the same as that described in previous articles.

To get the culture samples from patients’ pus or wounds, we used sterile syringes to aspirate the pus and discharge from abscesses or wounds with pus formation. The culture samples were collected during the procedure if a patient accepted debridement.

If the aspiration procedure was difficult, our staff would use sterile cotton swabs to stick the wound discharge. The aerobic culture and anaerobic culture would both be collected. 

Before these procedures, our staff wound sterilize the wound surface and skin around the bite wounds with 75% rubbing alcohol solution. After all procedures were completed, the culture samples would be sent to our central laboratory as soon as possible.

Wounds were defined as infected snakebite wounds if the clinical conditions were consistent with the following criteria: (1) observation of microorganisms in wound or pus culture reports, (2) diagnosis of cellulitis, abscess, or necrotizing fasciitis in medical records, and (3) experience of wound debridement by a surgeon.

Patients with *N. atra* bites who developed wound necrosis were defined as having wound infection. However, these patients’ clinical variables were not used to validate the BITE and Cobra BITE scores. We only used their bacteriology results for the selection of antibiotics. To describe bacteriology findings in the culture report, we defined polymicrobial infection as the discovery of two or more microbial species in one infected wound.

### 5.3. Statistical Analysis and Validation of the BITE and Cobra BITE Scores

Categorical variables are described as the frequency and percentage, and continuous variables are described as the mean and standard error of the mean. Univariate analysis was performed using Student’s *t* test (for numerical variables) and the chi-square test (for categorical variables). A *p* value of <0.1 was considered statistically significant. We used scoring systems described in the BITE and Cobra BITE studies to evaluate infection risk. Then, we calculated ROC curves and AUC for the validation cohort. Finally, we compared the AUC of the derivation set (BITE and Cobra BITE studies) and the validation set using the DeLong test. A *p* value of <0.05 was regarded as statistically significant. Statistical analyses were performed using SAS version 9.2 (SAS Institute, Cary, NC, USA).

## Figures and Tables

**Figure 1 toxins-15-00078-f001:**
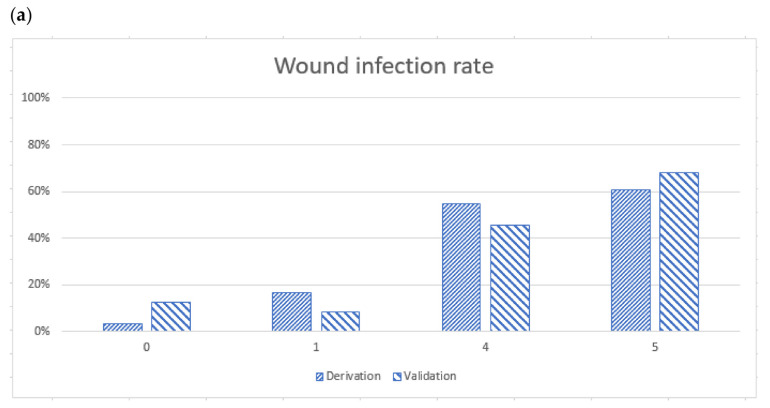
(**a**) Wound infection rates determined using the BITE scores, and the X axe meant the BITE score points (**b**) Receiver operator characteristic curves for derivation and validation sets. The derivation set was the previous BITE study, and the area under the curve (AUC) was 0.8391. The AUC of the validation cohort was 0.7814. Both the derivation and validation AUC were >0.5. We suggest the administration of antibiotics to patients with a BITE score of 4 or 5 points.

**Figure 2 toxins-15-00078-f002:**
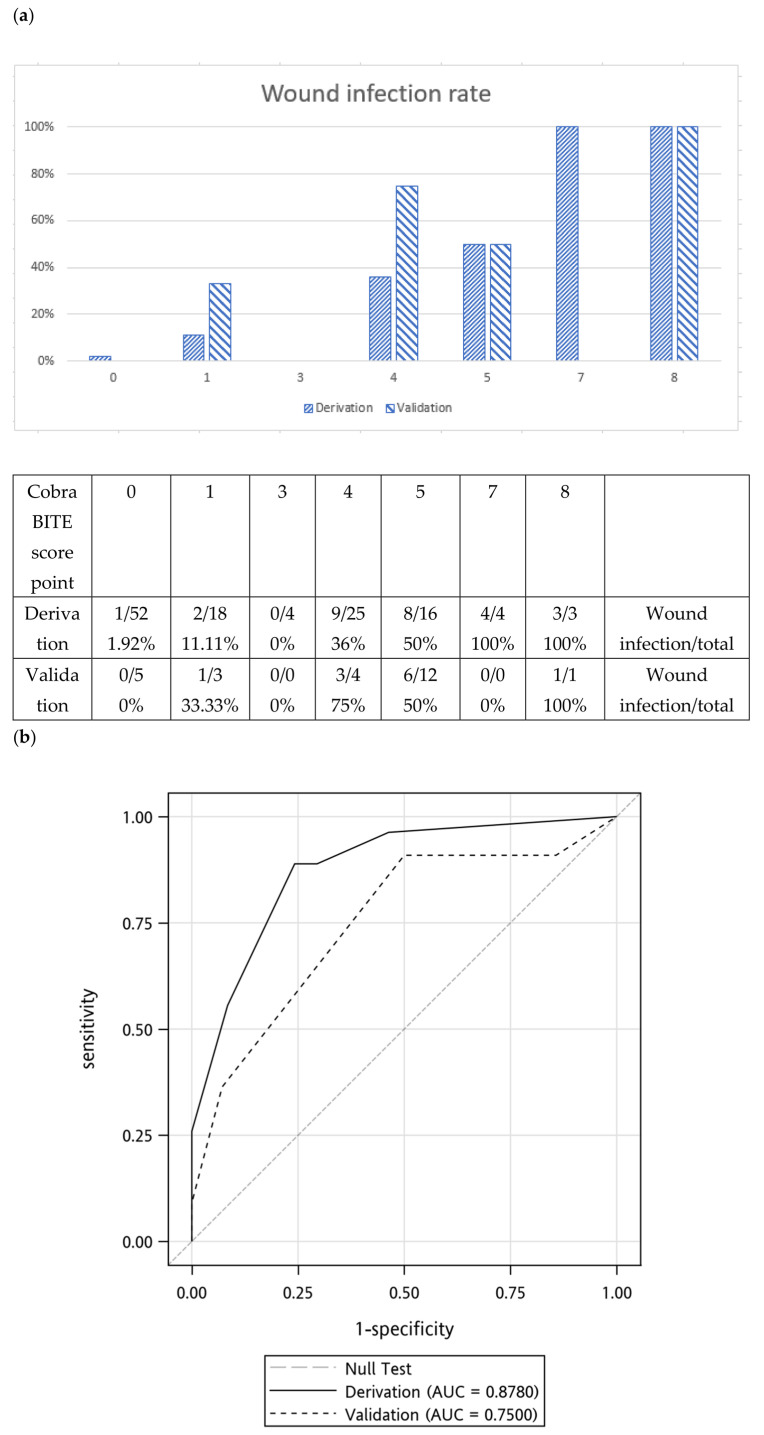
(**a**) Wound infection rates determined using the Cobra BITE scoring system and the X axe meant the Cobra BITE score points (**b**) Receiver operator characteristic curves for derivation and validation sets. The derivation set was the previous Cobra BITE study, and the area under the curve (AUC) was 0.8780. The AUC of the validation cohort was 0.7500. Both the derivation and validation AUC were >0.5. Antibiotics should be given if the Cobra BITE score is ≥5.

**Figure 3 toxins-15-00078-f003:**
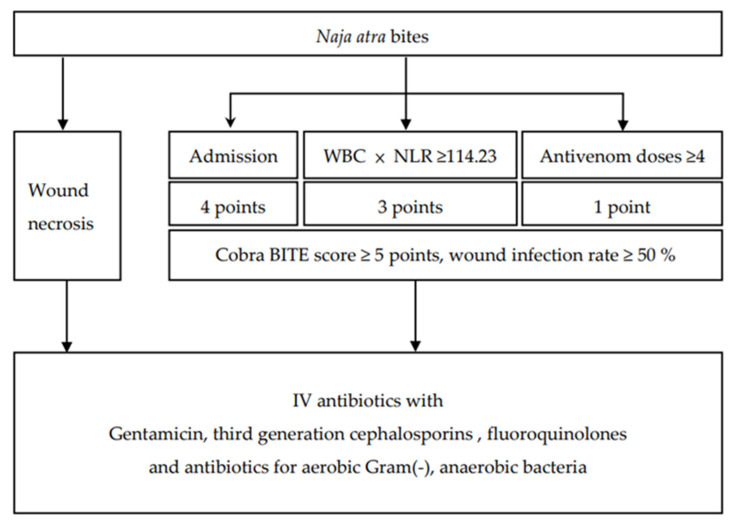
The Cobra BITE score and the suggested antibiotics for patients bitten by *Naja atra*.

**Figure 4 toxins-15-00078-f004:**
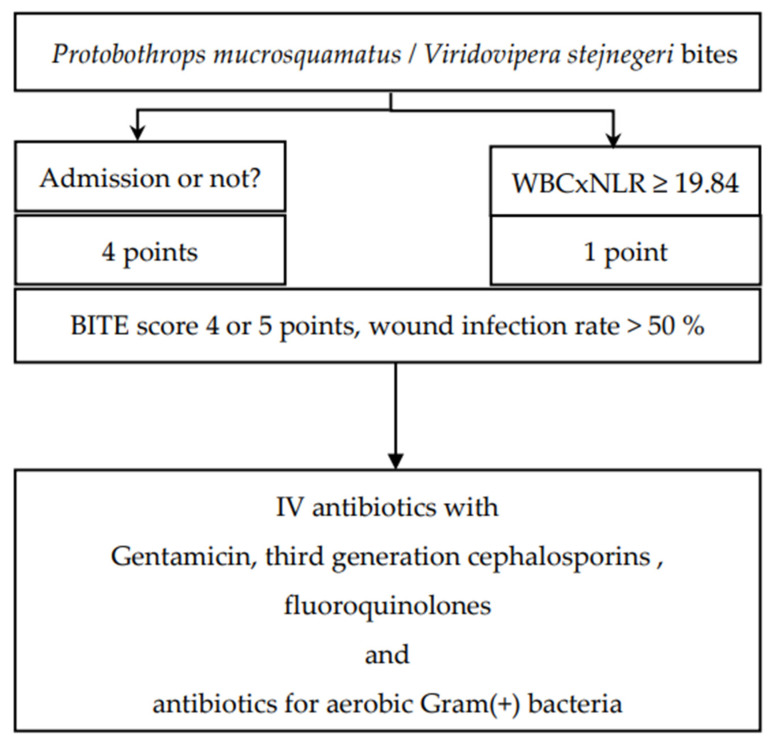
The BITE score and suggested antibiotics for patients bitten by *Protobothrops mucrosquamatus* and *Viridovipera stejnegeri*.

**Table 1 toxins-15-00078-t001:** Patient characteristics of the validation cohorts and variables used in the validation sets of the BITE and Cobra BITE studies.

**BITE Score**			
	Derivation (*n* = 726)	Validation (*n* = 95)	*p*
Age, mean (SD)	52.07(17.55)	59.05(16.03)	<0.01
Male, %	506(69.70)	64(67.37)	0.64
WBC (1000/μL) × NLR ^※^ ≥ 19.84	196(40.83)	40(42.55)	0.76
Admission	230(31.68)	51(53.68)	<0.01
Wound infection rate	163(22.45)	35(36.84)	<0.01
**Cobra BITE score**			
	Derivation (*n* = 172)	Validation (*n* = 25)	*p*
Age, mean (SD)	48.94(17.43)	56.68(19.52)	0.04
Male, %	125(72.67)	20(80)	0.44
WBC × NLR ^※^ ≥114.23	12(9.84)	1(4)	0.92
Admission	51(29.65)	17(68)	<0.01
Antivenin dose ≥4 vials	43(25)	16(64)	<0.01
Wound infection rate	30(17.44)	11(44)	0.01

**^※^** NLR: neutrophil-lymphocyte ratio.

**Table 2 toxins-15-00078-t002:** Microorganisms found in the wound cultures of patients with *Protobothrops mucrosquamatus*, *Viridovipera stejnegeri*, and *Naja atra* bites ^%^.

Microorganism	*Naja atra* Bites, *n*	*Protobothrops mucrosquamatus*/*Viridovipera stejnegeri* Bites, *n*
**Aerobic Gram-positive bacteria**		
*Bacillus*	1	-
*Coag_staphylococcus*	3	1 + 1
*Enterococcus_faecalis*	13 + 4 ^#^	3
*Corynebacterium jeikeium*	-	2
*Corynebacterium spp*	-	1
*Staphylococcus saprophyticus*	-	1
*Viridans streptococcus*	-	1
*Staph_aureus*	1	2
**Aerobic Gram-negative bacteria**		
*Acineto_baumannii*	1	-
*Enterobacteriaceae*	1	-
*Acinetobacter_sp*	1	-
*Shewanella_algae*	2	-
*Morganella_morganii*	16 + 2	3
*Aeromonas hydrophila*	-	1
*Stenotrophomonas maltophilia*	-	1
*Klebsiella pneumoniae*	1	-
**Anaerobic bacteria**		
*Peptostreptococcus micros*		1
*Proteus vulgaris*	3	-
*Citrob_freundii*	1	-
*B_fragilis*	2	1
*B_thetaiotaomicron*	1	-
*Serratia_marcescens*	2	-
*Providencia_rettgeri*	1	-
*Enterobacter cloacae*		1 + 1
*Enterobacter cloacae CR strain*		1
**Others**		
Yeast_like	1	-

^#^ n, 13 + 4 = cultured 13 times of the same bacteria species in the derivation cohort and 4 times of the same bacteria species cultured in validation cohorts. ^%^ This table combines both derivation and validation cohort outcomes.

**Table 3 toxins-15-00078-t003:** Use of empirical antibiotics and antibiotic susceptibility.

Cultured Bacteria	Empirical Antibiotics *	Bacteria Resistant Antibiotics
**Taiwan Cobra infected wound**		
*Morganella morganii*	AMC	CXM, CFZ, SAM
*Proteus vulgaris*	OX, FLO	CXM, CFZ
*Enterococcus.faecalis*	OX, FLO	--
*Morganella morganii*	AMC	CXM, CFZ
*Proteus vulgaris*	AMC	CXM, CFZ
*Enterococcus faecalis*	OX, CRO	--
*Proteus vulgaris*	OX, CRO	CXM, CFZ
*Enterococcus faecalis*	AMC, CXM, CC	--
*Enterobacteriaceae*	AMC, CXM, CC	CFZ
*Kleb.pneumoniae*	AMC, CXM, CC	--
*Kleb.pneumoniae*	AMC, CXM, CC	--
*Enterococcus. Faecalis*	AMC, CXM, CC	--
***Protobothrops mucrosquamatus* and *Viridovipera stejnegeri* infected wound**		
*Enterobacter cloacae complex*	CFZ, AMC	CXM, CFZ, SAM
*Coagulase(-) staphylococcus*	CFZ, AMC	--

* Antibiotics used before wound culture reports were listed according to the time sequence. CFZ, cefazolin; CXM, cefuroxime; SAM, ampicillin with sulbactam; AMC, Amoxicillin with clavulanic acid;CC, clindamycin; OX, oxacillin; FLO, flomoxef.

## Data Availability

Not applicable.

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
