# Peer review of "Wound Infection of Snakebite from Venomous Protobothrops mucrosquamatus, Viridovipera stejnegeri and Naja atra in Taiwan: Validation of BITE and Cobra BITE Scoring Systems and their Bacteriological Differences in Wound Cultures"

_toxins, 2023, doi:10.3390/toxins15010078_

Round 1

Reviewer 1 Report

The manuscript compares snakes' treatment/infection scoring systems (BITE). The number of patients analyzed is statistically significant, and the statistical analysis methodology is adequate. The manuscript deserves publication, however, some small corrections are necessary beforehand:

Major points:

-Remove the decimal .00 from the graphics in Fig.1 and Fig. 2, leaving only 0-80%, as it is statistically insignificant at this decimal point sensitivity.

-The discussion is being presented in a segmented way, which is unusual for this Journal, making it difficult for the reader to interpret the work as a whole.

-The supplementary tables were not inserted in a repository.

-It is important to mention (company, state, country) the materials and antibiotics used.

-Insert the concentration of antibiotics used in the treatment in the text or table

Reviewer 2 Report

First--Congratulations on this long-term work that has both regional and international significance.  This is the first really convincing study I am aware of about who needs antibiotics, which ones and how to make a decision. This work will provide a model for others to build from in their local environments or to adapt. While at UC Berkeley, the great & wise Taiwanese chemist YT Lee frequently said, "if it was easy, it would have been done"--and this work was obviously not easy, but now it nearly done. Reading this paper made me think of him and his interest in snakes and snake iconography of southern Taiwan. 

My comments about the readability of the manuscript are minor. My only suggestions are the following: 

1. Table 1: Suggest capitalizing the first word of each new line. It is easier to discriminate when there is a new row and to track (e.g. Age rather than age)

2. I suggest putting/describing the specifics of the scoring system (currently figure 3 and 4) much earlier --e.g. between Introduction and results. I think the authors are very methodical about proving their case, but as a clinician reader I am more inclined to look at the scoring system, decide if it is something I would use (yes!) and then read to see if I am convinced of the evidence. To that end: I recommend a "bottom line up front" approach. Convince me I should read the paper. This comment is entirely subjective and I believe the paper can be accepted as it is written. I am merely commenting on my own behavior as a practicing physician and one who is probably quite average in personality and disposition toward reading medical literature.  

I appreciated inclusion of the supplemental data, too. 

Wonderful work. I hope to see others build on it and can imagine its adaptation to other regions such as India and subsaharan Africa. Congratulations. 

Reviewer 3 Report

The manuscript describes validating two scoring systems to prescribe antibiotics for treating infections of local necrosis induced by snakebites.

The results are clear, and the data support the conclusions.

The data on the culture of the oral cavity of snakes and wounds after snakebite is not a novelty, but illustrates the problem of wound infection and some difficulties in treating it.

This publication can interest clinicians in managing snakebites and using antibiotics to complement treatment after using antivenoms.

Reviewer 4 Report

This paper describes the verification of BITE rating of snake bites. At the same time, the risk of microbial infection caused by venomous snake bite was studied, and suggestions on prevention and treatment of infection were put forward. It has certain guiding significance for the treatment of snakebite. However, the paper still has the following problems: 

1. The annotations in Figures 1 and 2 are too simple. It is suggested to supplement the description of coordinates and other necessary information.

2. In Table 2, are there multiple microorganisms in the same patient? The author is requested to verify whether these data will overlap.

3. The microorganisms bitten by venomous snakes may come from the skin, mouth, or even the venomous glands of venomous snakes. It may also come from the patient's own skin or environment. Can we discuss the source of microbial infection?

4. Microbial culture conditions are very important for the identification of microorganisms (microbiology). Although the author has described the cultivation methods in other papers, in view of the importance of the cultivation and identification methods for this paper, it is recommended to supplement the corresponding methods in this paper.

5. It is suggested to add some full names of, such as WBC.

Reviewer 5 Report

In this manuscript the authors describe bacterial wound infection of patients bitten by three different snakes existing in Taiwan. Differently from most  snake species, the venomous secretion of the studied snakes produces bacterial wound infection.  Bacteriological studies of new bitten  patients were compared with data obtained in their previous work. The work shows scientific  soundness, however some points must be clarified:

1. the concept of derivation and that of validation must be better explained in the text;

2. Materials and Methods chapter is poorly described. It requires explanation of the methodology and materials used to allow the reader to better understand what kind of study was designed and the material used in order to justify the experiment to be made; 

3. An abbreviation section would facilitate the reading of the text; 

4. The figures requires legends. For instance: Figure 1a: What does it means the numbers 0 to 5 in the X axe? Figure 1b: it information is not clear. 

Round 2

Reviewer 1 Report

The authors responded adequately to the questions and made the requested changes. Small adaptations in English are still necessary, however, now the paper can be accepted for publication.